# The Multi-Channel System of the Vietnamese Mekong Delta: Impacts on the Flow Dynamics under Relative Sea-Level Rise Scenarios

Hoang-Anh Le [1,2,*], Thong Nguyen [1], Nicolas Gratiot [1,3], Eric Deleersnijder [4] and Sandra Soares-Frazão [2]

[1] Asian Research Center on Water (CARE-Rescif), Ho Chi Minh City University of Technology, Block B7, 268 Ly Thuong Kiet Street, District 10, Ho Chi Minh City 700000, Vietnam; nthong56@yahoo.fr (T.N.); nicolas.gratiot@ird.fr (N.G.)

[2] Institute of Mechanics, Materials and Civil Engineering (IMMC), Université catholique de Louvain, Place du Levant 1, B-1348 Louvain-la-Neuve, Belgium; sandra.soares-frazao@uclouvain.be

[3] CNRS, IRD, Grenoble INP, IGE, Université Grenoble Alpes, F-38000 Grenoble, France

[4] Institute of Mechanics, Materials and Civil Engineering (IMMC) & Earth and Life Institute (ELI), Université Catholique de Louvain, 4 Avenue Georges Lemaître, B-1348 Louvain-la-Neuve, Belgium; eric.deleersnijder@uclouvain.be

\* Correspondence: anhle.hn85@gmail.com

**Abstract:** The Mekong Delta has the world's third-largest surface area. It plays an indisputable role in the economy and livelihoods of Vietnam and Cambodia, with repercussions at regional and global scales. During recent decades, the Vietnamese part of the Mekong Delta underwent profound human interventions (construction of dykes and multi-channel networks), which modified the hydrodynamic regime, especially cycles of field submersion. In this study, we first applied a full 2D numerical hydraulic model, TELEMAC-2D, to examine the effects of the complex channel and river networks on the spatial and temporal distribution of the flow in the 40,000 km$^2$ of the Vietnamese Mekong Delta. Then, two scenarios of relative sea-level rise in 2050 and 2100 were implemented to simulate the future patterns of water fluxes in the delta. The results show that dykes and multi-channel networks would reduce the inundation area by 36% and lessen the peak water level by 15% and the discharge over the floodplains by 24%. Despite this protection, under a relative sea-level rise of 30 cm and 100 cm, the maximum flooded area could occupy about 69% and 85% of the whole delta in 2050 and 2100, respectively.

**Keywords:** Vietnamese Mekong Delta; TELEMAC-2D; hydrodynamics; channel system; sea-level rise

## 1. Introduction

River deltas have been of fundamental importance to civilization for thousands of years [1,2] because of their fertile flat lands, abundant freshwater for living and agricultural practices, fishing grounds and suitability for fluvial transport [3]. However, the natural state of these deltas is now being modified dramatically by ongoing human interventions [4]. Unsurprisingly, this holds true for the Mekong Delta [5].

The Mekong Delta covers an area of 55,000 km$^2$ in Cambodia and Vietnam [6,7], with an area of 26% in the former country and 74% in the latter [8]. This delta is at the core of various interdependent and endangered economic sectors, encompassing agriculture, fishery and forestry [9,10]. The Cambodian portion of the delta exhibits significant differences from Vietnam's. In Cambodia, the floodplain has witnessed little anthropogenic influences and can be considered smoothly impacted, with only a few control structures, whereas the Vietnamese part is under large regulations by a huge system of navigation and irrigation channels, sluice gates, pumps and extensive dyke systems. These systems have considerably altered the natural hydrodynamics and sediment transport [11,12]. Recently, researchers have paid more attention to the whole domain through extensive monitoring

networks [13,14] and satellite observations [7,15] and by applying 1D [16–18], semi/quasi 2D [19] or 1D–2D coupled flow simulation models [8,12]. Other studies concentrated on the impacts of dyke systems on hydrodynamics [20,21]. Recently, many studies have also highlighted the phenomenon of sea-level rise and subsidence in the Mekong Delta region [5,22,23] and addressed the root causes of subsidence [5,24]. Researchers have warned that subsidence along the VMD coastline has increased rapidly since 2005 and is not only up to four times higher than the average sea-level rise but also varies along the entire estuary [25]. A study by [26] provides an estimation of the spatial and temporal scale of the relative sea-level rise (RSLR) on deltas and coastal communities all over the world. They confirmed that the low rates of land subsidence and sea-level rise are key to the long-term sustainability of coastal deltas, but the construction of new dams and reservoirs is estimated to contribute on the order of an additional 1 mm/y of the RSLR. Even so, only a few research works have comprehensively assessed the flow dynamics at a large scale. A typical research revealing riverbed morphological evolution was conducted by [27], in which they simulated riverbed changes in Vam Nao conjunction, which is tidal-influenced and free from sand mining, and in which they experienced riverbank failures. It must be stressed that overland flow and water exchange between irrigation compartments influence delta conditions [28] in ways that cannot be simulated by 1D models.

The present study utilizes a two-dimensional numerical model, TELEMAC-2D (http://www.opentelemac.org/, accessed on 18 January 2020) [29], to evaluate the flux of water in a fully spatialized domain, i.e., to examine the effects of the multi-channel network on spatial and temporal distribution of flow in this large-scale domain. To evaluate the alterations by both natural conditions and development activities, two simulation scenarios are taken into consideration: (i) the delta in a natural state, without channels and dykes, and (ii) the delta in its current conditions. Additionally, the Vietnamese Mekong Delta (VMD), like other large lowland drainage systems around the world, is also affected by natural drivers, e.g., fluvial sediment reduction [30], salinization [31,32], coastal erosion [33,34], land subsidence [20,22,35] and climate change and especially global sea-level rise [8,23,36–38]. Based on the report on climate change and sea-level rise scenarios for Vietnam, which was established by the Ministry of Natural Resources and Environment [39], four additional scenarios are taken into account in this study to assess the impacts of RSLR on hydrodynamics in the VMD by 2050 and 2100. Based on the six studied scenarios, the full 2D model will be used for the following:

- Simulating the spatial and temporal distribution of the inundation processes;
- Reevaluating impacts of the multi-channel network on the flow dynamics;
- Forecasting the future patterns of water flux under different scenarios of RSLR in 2050 and 2100.

This paper is organized as follows: Section 2 introduces the study area; Section 3 presents the model setup; in Section 4, model calibration and validation are run; Section 5 shows the simulation results for the six scenarios; Section 6 discusses the impacts of the multi-channel network and RSLR in the Vietnamese Mekong Delta; and, finally, conclusions about the impacts of the multi-channel system on flow and sediment dynamics in the VMD are drawn.

## 2. Study Area

The Mekong Delta was formed by the deposition of sediment from the Mekong River over thousands of years [33,40,41], resulting in a total area of 55,000 km$^2$ [6,7], and is the third largest delta in the world [33]. The Vietnam Mekong Delta (VMD) is located in the southernmost part of the Mekong Delta. It covers an area of approximately 40,000 km$^2$ [38], accounting for 12% of Vietnam's territory area, and it is the homeland of 18 million people [33,42]. It plays an indisputable role in the Vietnamese economy and local residential livelihoods as three-quarters of this region is utilized for agricultural production [43]. About 56%, 50% and 70% of Vietnam's rice, fish and fruit production originates from the VMD.

The delta is one of the lowest-lying tropical areas [22], with a mean topography elevation of 0.82 m (above sea level, a.s.l.). The elevation of the central part ranges from 1.0 to 1.5 m, whereas coastal areas have a lower elevation of 0.3–0.7 m [40]. The delta's hydrodynamics is driven by two major tributaries [17], the Tien River (the Mekong River) and the Hau River (the Bassac River), which drain into the East Sea of Vietnam through eight estuaries (Figure 1a). Based on their characteristics and functions, the delta is commonly divided into four sub-basins: the Long Xuyen Quadrangle (LXQ, number 1 in Figure 1b), the Plain of Reeds (PoR, number 4 in Figure 1b), the area between the Tien River and Hau River (number 2 in Figure 1b) and the Ca Mau Peninsula (number 3 in Figure 1b).

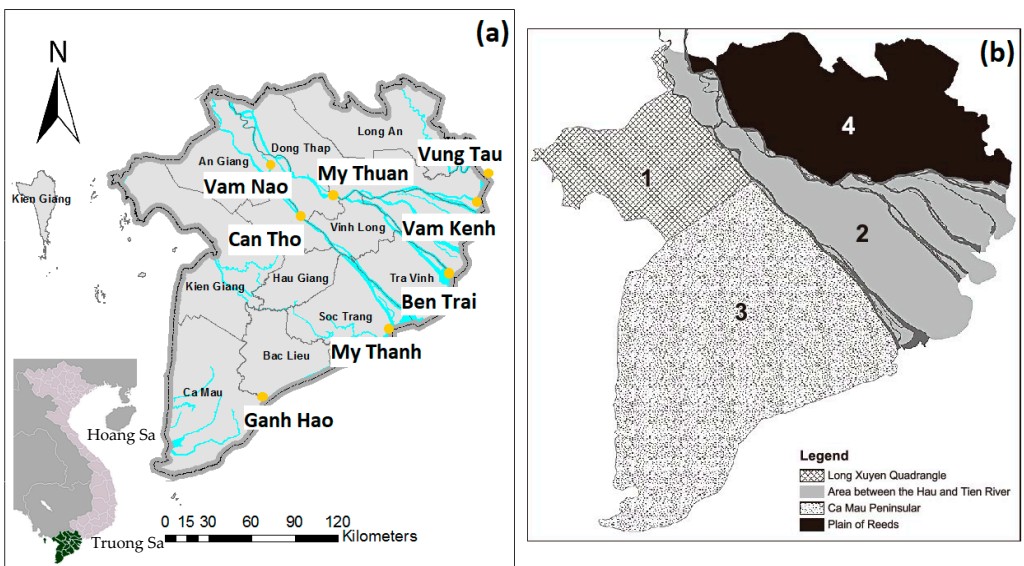

**Figure 1.** (**a**) Hydrological stations (orange dots) used for the calibration and validation of the model. Vam Nao station is located in the conjunction of the Tien and Hau Rivers. My Thuan and Can Tho stations are on the Tien River and Hau River, respectively. Vung Tau, Vam Kanh, Ben Trai, My Thanh and Ganh Hao stations are representative stations distributed along the coast. (**b**) Sub-regions of the VMD: (1) Long Xuyen Quadrangle, (2) Area between the Hau and Tien River, (3) Ca Mau Peninsular and (4) Plain of Reeds [44].

A distinct feature of the VMD is the intensive management of the multi-channel and dyke systems in order to mitigate floods and saltwater intrusion and to optimize agricultural activities and fluvial transportation [17,21]. Nowadays, these structures comprise 7000 km of main channels, 4000 km of secondary on-farm channel systems, 193 spillways, 409 reservoirs, 528 junctions, 29 sluices and 749 compartments [37]. The system is concentrated on flood-prone areas to efficiently drain out floodwater from the LXQ and the PoR to the Gulf of Thailand (the West Sea of Vietnam) and to the Vam Co River before debouching into the East Sea of Vietnam [14]. The channel system in the VMD is fully interconnected, without separation between irrigation channels and drainage ones [42]. The main channels take water directly from the Tien River and the Hau River. They are 70–100 m in width and 3–5 m in depth. Compartments between dykes are connected to the main channels by a network of secondary channels, which are 30–50 m in width and 2–3 m in depth. Large parts of the VMD are controlled by sluice gates and pumps, which are managed by local authorities [11].

Together with the channel network, the dyke systems have been expanded since 1975 [45], when the demand for food increased sharply after the Vietnam reunification [45]. After the devastating flood of 2000, the dyke systems were especially reinforced in order to maintain agricultural cultivation during flood seasons. The dyke systems comprise low- and high-ring dykes. The low dyke rings, with an average crest level of about 2.0–2.5 m a.s.l. [13], aim to protect paddy fields against the early flood peak from mid-July

to mid-August so that farmers can cultivate two rice crops per year [19]. The high dyke rings were designed with an average crest level of about 4.0–4.5 m a.s.l. [11], approximately 0.5 m above the flood peak of the year 2000, and are located mainly along the banks of the Tien River and Hau River [13]. The high dyke rings aim at protecting rice fields during the whole year and regulating climate change-related floods. Thus, they can facilitate the cultivation of three crops per year in An Giang and Dong Thap provinces (Figures 1a and 2). However, this infrastructure system has been the subject of much debate in the Vietnamese scientific communities and abroad because it causes an alteration of flood pulse and sedimentation in the floodplain, posing several socio-economic issues [46].

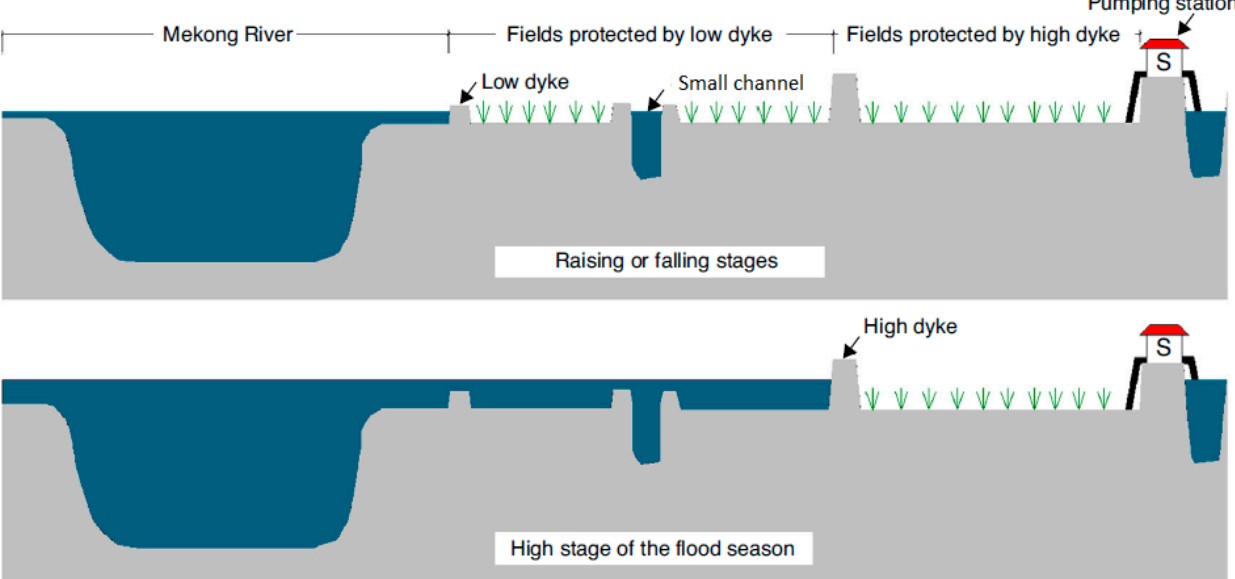

**Figure 2.** Simple river and floodplain cross-sections highlighting the effect of flood prevention infrastructure in the floodplains [21,47].

Located in the North Pacific monsoon climate region, the VMD experiences tropical monsoon characteristics, with two separate seasons per year. The wet season normally lasts from June to November, whereas the dry season lasts from December to May [12]. The precipitation in the wet season contributes to approximately 85% of the annual rainfall and leads to the flooding of large areas in the delta [48]. In addition, the region is impacted by tropical cyclones [30], causing difficulties in predicting flood behaviors and inundation [4]. These flooding dynamics, however, have also contributed to the area's highly fertile alluviums and fish productivity for centuries [49,50].

The total annual water discharge of the VMD experiences about 500 billion m$^3$ (84.4% from the upstream and 15.6% from the regional rainfall). The flow is the highest during the flood season, particularly in September and October, when it can reach up to 25,500 m$^3$/s. The flow during the dry season is rather low, with a mean discharge no larger than 6000 m$^3$/s, which leads to saline intrusion in some river mouths [40].

The VMD estuaries experience semidiurnal tides (M2, S2, N2, K2) originating from the East Sea of Vietnam with amplitudes of 1–3.5 m and diurnal tides (O1 and K1) from the West Sea of Vietnam with amplitudes of 0.8–1 m [8]. The tides also influence the hydrodynamic condition in the VMD, as identified by [14] through bed sample analysis and by [34] through coupled numerical and in situ/satellite observations.

## 3. Model Setup

### 3.1. TELEMAC 2D

The TELEMAC-MASCARET system (http://www.telemacsystem.com, accessed on 18 January 2020) [29], which was introduced by the National Hydraulics and Environmental Laboratory, a part of the R&D group of Electricité de France (EDF), is a numerical modelling

system that focuses on environmental processes in free surface transient flows. The primary purpose of TELEMAC is to simulate the flow dynamics in a waterbody via the solution of the shallow water equations [51]. Ref. [52] applied TELEMAC-2D for the hydrodynamic modelling of the tide propagation in the Guayas delta of Ecuador, a tropical delta, while [53] set up a hydrodynamic model based on the TELEMAC 2D to implement flood extent simulation in the Inner Niger Delta, Mali, West Africa. Using a finite-element method, TELEMAC-2D solves the continuity and momentum equations on an unstructured mesh made up of triangles. The governing equations read as follows:

$$\frac{\partial h}{\partial t} + \vec{u}.\vec{\nabla}h + h\,div(\vec{u}) = S_h \tag{1}$$

$$\frac{\partial u}{\partial t} + \vec{u}.\vec{\nabla}u = -g\frac{\partial Z}{\partial x} + S_x + \frac{1}{h}div(h\vartheta_t\vec{\nabla}u) \tag{2}$$

$$\frac{\partial v}{\partial t} + \vec{u}.\vec{\nabla}v = -g\frac{\partial Z}{\partial y} + S_y + \frac{1}{h}div(h\vartheta_t\vec{\nabla}v) \tag{3}$$

where $h$ is the water depth (m); $u$ and $v$ are depth-averaged horizontal velocity components (m/s), with $\vec{u} = (u,v)$; $g$ is the gravitational acceleration (m/s$^2$); $Z$ is the free surface elevation (m), positive upward; $t$ is time (s); $x$ and $y$ are horizontal cartesian coordinates (m); $S_h$ is the source or sink of water (m/s); $S_x$ and $S_y$ encompass the Coriolis force as well as the bottom and surface stress terms (m/s$^2$); $\vartheta_t$ is the horizontal kinematic viscosity (m$^2$/s) and $\nabla$ is the (horizontal) del operator.

### 3.2. Data Utilization

All data used as inputs for the model were obtained from official sources, summarized in Table 1, e.g., HoChiMinh City University of Technology, Vietnam, the Southern Institute for Water Resources Research and Vietnam MRC. The Lower Mekong Coastal Delta Zone project (LMCDZ) was funded by the European Union (EU) and the French Development Agency (AFD) and was coordinated by institutes from Vietnam (SIWRR) and France (IRD, hosted by the CARE research lab in HCMC) and about 10 international experts from France, Germany, the Netherlands and Vietnam.

**Table 1.** Data sources.

| Data Type | Frequency | Data Source |
|---|---|---|
| River network and channel cross-sections | Surveyed from 1995 to 2000 with updates between 2005 and 2010 | HoChiMinh city University of Technology, Vietnam |
| Hydraulic infrastructure operations | Hydraulic infrastructure embedded in the model are based on official regulations for 2010–2011 | Southern Institute for Water Resources Research, Vietnam MRC |
| Hydrological data | Can Tho and My Thuan (hourly Q and H in 2010 and 2011) | Lower Mekong Coastal Delta Zone project |
| | Tan Chau and Chau Doc (hourly Q and H) in 2010 and 2011 Vung Tau, Ben Trai, Ganh Hao, Vam Kenh (hourly H) in 2010 and 2011 | National Center for Meteo-Hydrological forecasting, Vietnam Vietnam–German University |
| Offshore wind | The hourly wind data at 10 m | NCEP NOAA |
| Offshore tidal constituents | Amplitude and phase of tidal constituents | TPXO 8.0 |

Figure 1a shows the location of hydrological stations used for model validation. Hourly data allows for capturing tidal influences and investigating the phase coherence between different gauges.

- For the estuarine areas (Soai Rap, CuaTieu, Cua Dai and Ham Luong) and the coastal areas of Go Cong, Can Gio and the Ganh Rai Gulf, the topographic data were extracted from the surveying reconnaissance of a 1/5000 scale topographic plane, in the years 2008, 2009 and 2010, under the framework of the Baseline Survey Project implemented by Vietnam's Southern Institute of Water Resources Research (SIWRR) and the Institute of Coastal and Offshore Engineering (ICOE) as well as survey work package of the Lower Mekong Coastal Delta Zone project (LMCDZ).
- For coastal areas from HCMC to Kien Giang, the topographic data were extracted from the map (scale of 1/100,000) published by the Navy in 1982.
- The topography in other areas of the sea was extracted from the SRTM30_PLUSV6.0 database from the Scripps Institution of Oceanography, Californian University, USA. This is a dataset with a $30'' \times 30''$ resolution, constructed from the satellite-gravity model, in which the gravity-to-topography ratios are corrected by 298 million ADCP depth points.

### 3.3. Computational Mesh

Figure 3 shows the domain and bathymetry of the VMD, which was developed from several sources and derived from the 100 m $\times$ 100 m grid resolution Digital Elevation Map (DEM), with reference to the Hondau datum (the Vietnamese official benchmark system, identical to mean sea level). In the upper part of the domain, the model is bounded by Tan Chau (the Tien River) and Chau Doc (the Hau River). In the estuarine area, the river tributaries drain into the sea and the model boundary is expanded by approximately 70–80 km offshore.

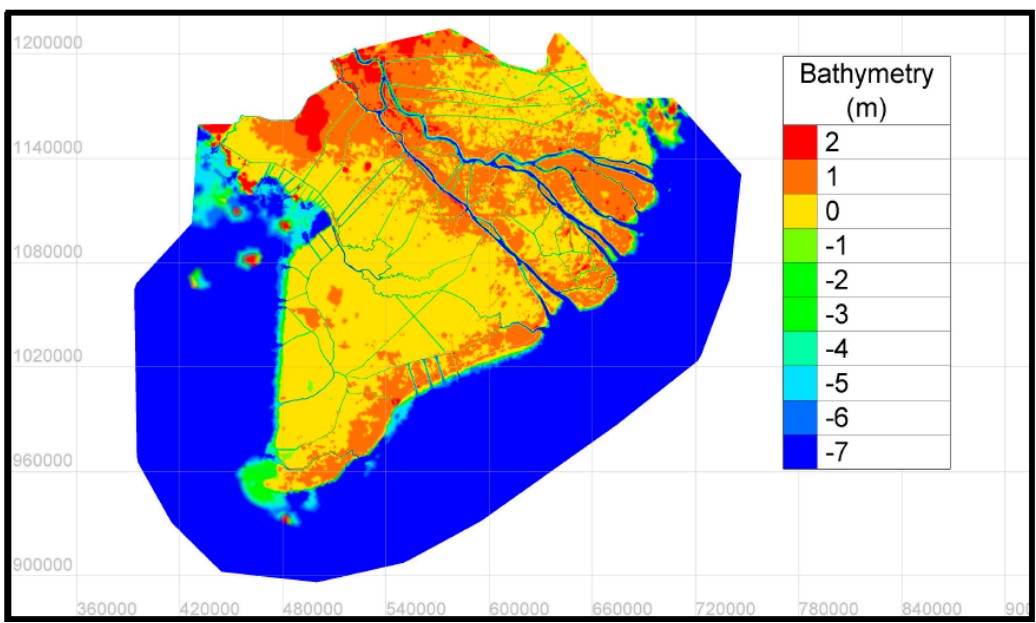

**Figure 3.** Bathymetry of the Vietnamese Mekong Delta.

The model was set up to represent the approximately real conditions of the VMD (Scenario S1). We modeled two main rivers (the Tien River and Hau River), eight estuaries and the primary and secondary channels. The tertiary and quaternary channels were neglected because the system connects with the two main rivers and drains directly into the sea. Roads and dykes are represented by high elevations in the bathymetry (see Figures 3 and 4).

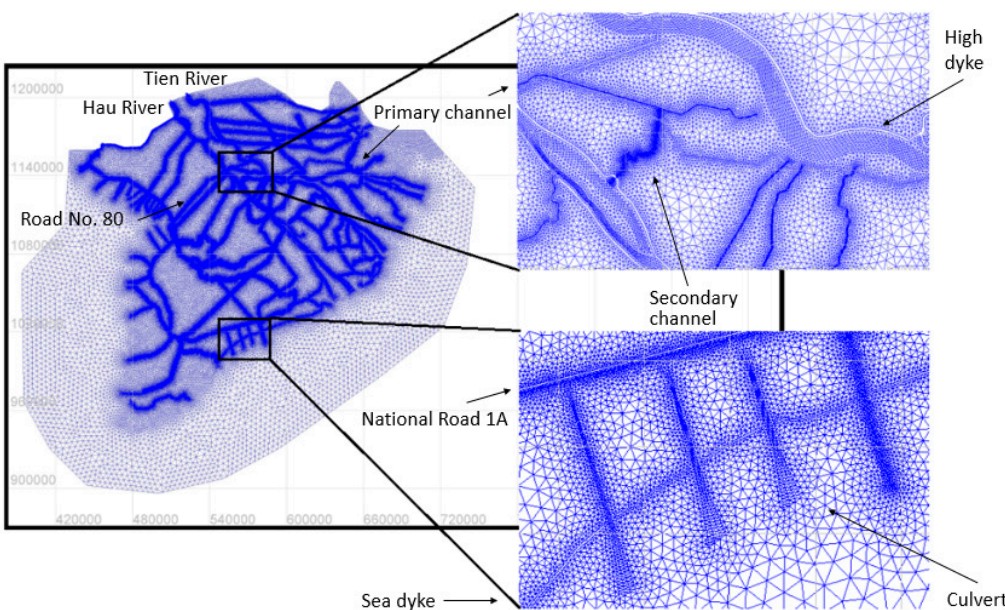

**Figure 4.** Computation mesh for Scenario S1 with main rivers, multi-channels, roads and culverts.

BlueKenue developed by the Canadian Hydraulics Centre of the National Research Council was utilized to generate the finite element mesh. The finite element computation mesh of triangular elements enabled the adaptation of the horizontal step in the study area. Triangular elements with small sizes of 80–100 m were used in the channels; a maximum size of 300 m was for the main rivers; a size of 300–2000 m was for the floodplain and the largest size of 4000 m was for the offshore settings. The total number of triangular elements is 524,097 with 268,010 nodes.

In order to assess the impacts of the numerous channels on the flow dynamics and sediment transport in the VMD, another computational mesh (see Figure 5) was generated with the two main rivers (the Tien River and Hau River) and eight estuaries, without the channel network (Scenario S2). The computational area is similar to that of Scenario S1. The mesh of this scenario is made up of 113,559 elements and 56,665 nodes. This second scenario (S2) can be viewed as representative of the delta situation prevailing before the construction of the dykes and channels, and a comparison of the results of Scenarios S1 and S2 will thus allow us to assess the impacts of these infrastructures.

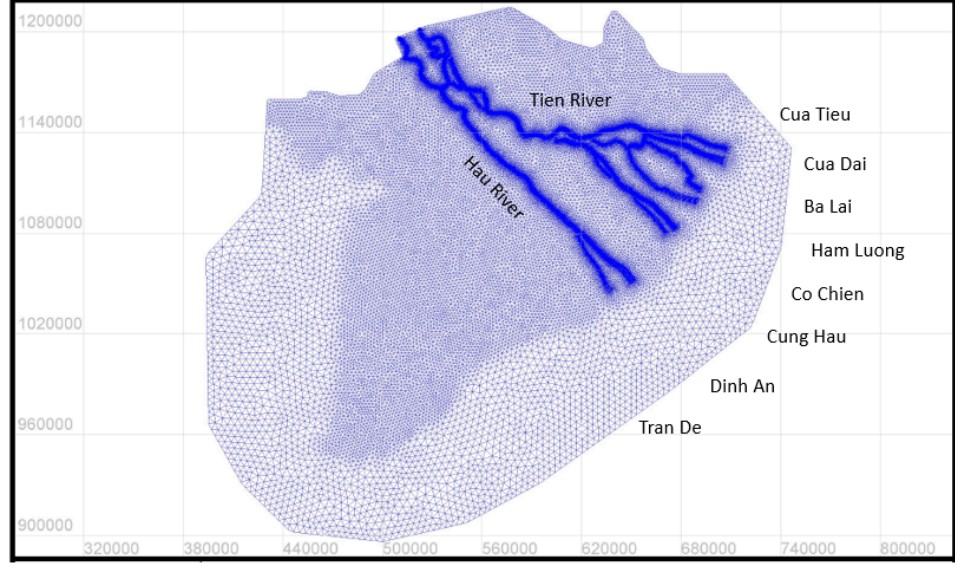

**Figure 5.** Computation mesh for Scenario S2 with the Tien and Hau Rivers.

*3.4. Parameters and Simulation Setup*

3.4.1. Boundaries

　　The upstream boundary conditions consist of measured hourly discharges imposed at Tan Chau (the Tien River) and Chau Doc (the Hau River). The tidal database of TPXO (https://www.tpxo.net/global) is applied as a real sea water level dataset in both the West and East Seas for the downstream boundaries. Coefficients to calibrate the sea level and the tidal range are also adjusted to transfer the information between a large-scale model and the boundaries of a local model [54].

3.4.2. Parameters of Hydrodynamic Simulation

　　The bed friction is parameterized by having recourse to the Nikuradse roughness length scale, $k_s$ [55], as shown in [56,57].

$$k_s \approx 3d_{50} \tag{4}$$

where $d_{50}$ is the median grain diameter (m).

　　The simulation uses a constant (horizontal) viscosity throughout the domain equal to $10^{-6}$ m$^2$/s. The default value incorporates both molecular viscosity and eddy viscosity in the constant viscosity model.

　　The wind is variable in time and space. The background wind data were derived from model results of the Climate Forecast System Reanalysis (CFSR) of NCEP/NOAA (https://www.ncep.noaa.gov/). Wind field results were obtained from "reanalysis" simulations, which include model validation with measured data from a global ocean monitoring station system (as a part of the Lower Mekong Delta Coastal Zone-LMCDZ project). Hourly wind data with a grid size of $0.312° \times 0.312°$ are used in the simulation. According to the introduction by the Institute of Oceanographic Sciences of the United Kingdom, the wind influence coefficient was set to $0.565 \times 10^{-6}$ because most of the wind velocity was <5 m/s (TELEMAC-2D User Manual) [29]. The atmospheric pressure was taken to be 105 Pa. The threshold depth of the wind was applied to avoid unphysical wind velocities with a constant value of 3 m.

　　Storm surge was not simulated in the study. Storms in the VMD occur mainly in November–December, coinciding with the phase of the highest freshwater runoff from the Mekong River, as well as with the phase of the highest seasonal sea-level setup. For example, Storm Linda affected the entire VMD on 2 November 1997 with a peak surge reaching a height of 1.5 m, coinciding with the highest freshwater discharge, highest tides and highest seasonal water level, and reaching up to 80 km inland. Linda caused severe damage and killed more than 3000 people [58]. However, the impact of Storm Durian in December 2006 was only on a local scale. The peak of its surge coincided with the lower low water level and caused no damage [25]. Hence, deep studies on storm surges could be implemented in the Mekong Delta.

　　The initial water elevation is 0 m. The simulation time-step was fixed at 20 s by trials and errors in order to ensure a balance between model stability and computational cost.

　　In order to assess the impacts of multi-channel networks on the flow dynamics in the VMD, two main scenarios (Scenario S1 and Scenario S2), as described above, were set up. However, as reported by [9,22,23,39], the VMD is one of the deltas that are the most vulnerable to climate change. RSLR is likely to be the main driver of climate change impacts in the delta. Thus, four additional scenarios were established corresponding to the two main scenarios with the projected relative sea-level rises of 30 cm in 2050 and 100 cm in 2100 [39]. These scenarios are summarized in Table 2.

　　In Scenario S1, the initial setup of the model was based on the current status of the VMD; the flood data is from the year 2011. The flood event in 2011 was one of two recent severe floods in Vietnam [38,59]. The measured daily discharge at Tan Chau and Chau Doc stations were used as the upstream boundary conditions, and the hourly sea levels at the East Sea and the West Sea were used as the downstream boundary conditions.

**Table 2.** Modelling scenarios.

| Scenario | Description |
|---|---|
| S1 | Two main rivers and the multi-channel network, as the current status of the VMD |
| S2 | Two main rivers without the multi-channel network, as representative of the delta situation prevailing before the construction of the dykes and channels |
| S3 | Same as Scenario S1 with a projected RSLR of 30 cm |
| S4 | Same as Scenario S2 with a projected RSLR of 30 cm |
| S5 | Same as Scenario S1 with a projected RSLR of 100 cm |
| S6 | Same as Scenario S2 with a projected RSLR of 100 cm |

In Scenario S2, the VMD consisted of only the Tien River and Hau River with eight estuaries, which correspond to the configuration of the delta before human intervention. The data for simulation are from the flood event in 2011. The measured daily discharge and the hourly measured sea levels were the same as in the previous scenario.

The model setup for Scenarios S3 and S4 is similar to Scenarios S1 and S2, respectively, with the same upstream discharges. However, the hourly sea levels at the downstream boundary were increased by 30 cm in both the East and West Seas, corresponding to the rising sea-level scenario in 2050. This value of 30 cm was extracted from the report on climate changes and sea-level rise scenarios for Vietnam, which was conducted by the Ministry of Natural Resources and Environment (MONRE) in 2016 [39].

Scenarios S5 and S6 have the same procedure as the previous scenarios with the sea-level rise of 100 cm at the downstream boundary condition, corresponding with the rising sea-level scenario in 2100.

## 4. Calibration and Validation

To assess the model performance during different periods and hydrological conditions, the periods of 10–19 September and 25 September–4 October 2011 were selected for the model calibration and validation because they are representative of the flood season in the VMD.

Data used for the model calibration and validation were extracted from several official sources and are presented in Table 1.

- Hourly data on discharge and water level in Can Tho and My Thuan stations in 2010 and 2011 were collected from the Lower Mekong Coastal Delta Zone project (LMCDZ).
- Hourly data on discharge and water level in Tan Chau and Chau Doc stations in 2010 and 2011 were collected from Vietnam's National Center for Meteo-Hydrological Forecasting and Vietnam–German University.
- Hourly data on the water level in Vung Tau, Ben Trai, Ganh Hao, Vam Kenh stations in 2010 and 2011 were collected from Vietnam's National Center for Meteo-Hydrological Forecasting and Vietnam–German University.

The model was calibrated by adjusting the Nikuradse roughness length scale in six areas (Table 3), as was the tidal coefficient parameter of −0.4 m and tidal range of 1.2 to transfer the information between a large-scale model (https://www.tpxo.net/global) and the boundaries of a local model [54].

**Table 3.** Nikuradse roughness for subdomains and the study area.

| No. | Subdomain | Nikuradse Roughness (m) | No. | Subdomain | Nikuradse Roughness (m) |
|---|---|---|---|---|---|
| **1** | Tien river | 0.1 | **4** | Co Chien estuary | 0.01 |
| **2** | Hau river | 0.12 | **5** | Ham Luong estuary | 0.01 |
| **3** | Vam Nao conjunction | 0.1 | **6** | The remaining area | 0.1 |

The water levels were evaluated at eight measurement stations (Figures 1a, 6 and 7 and Table 4), which are key stations with the highest data accuracy in the Mekong Delta. The quality of the match between simulated and observed water levels after calibration was evaluated by the Nash–Sutcliffe efficiency coefficient (NSE) [60], mean absolute error (MAE) and root mean square error (RMSE).

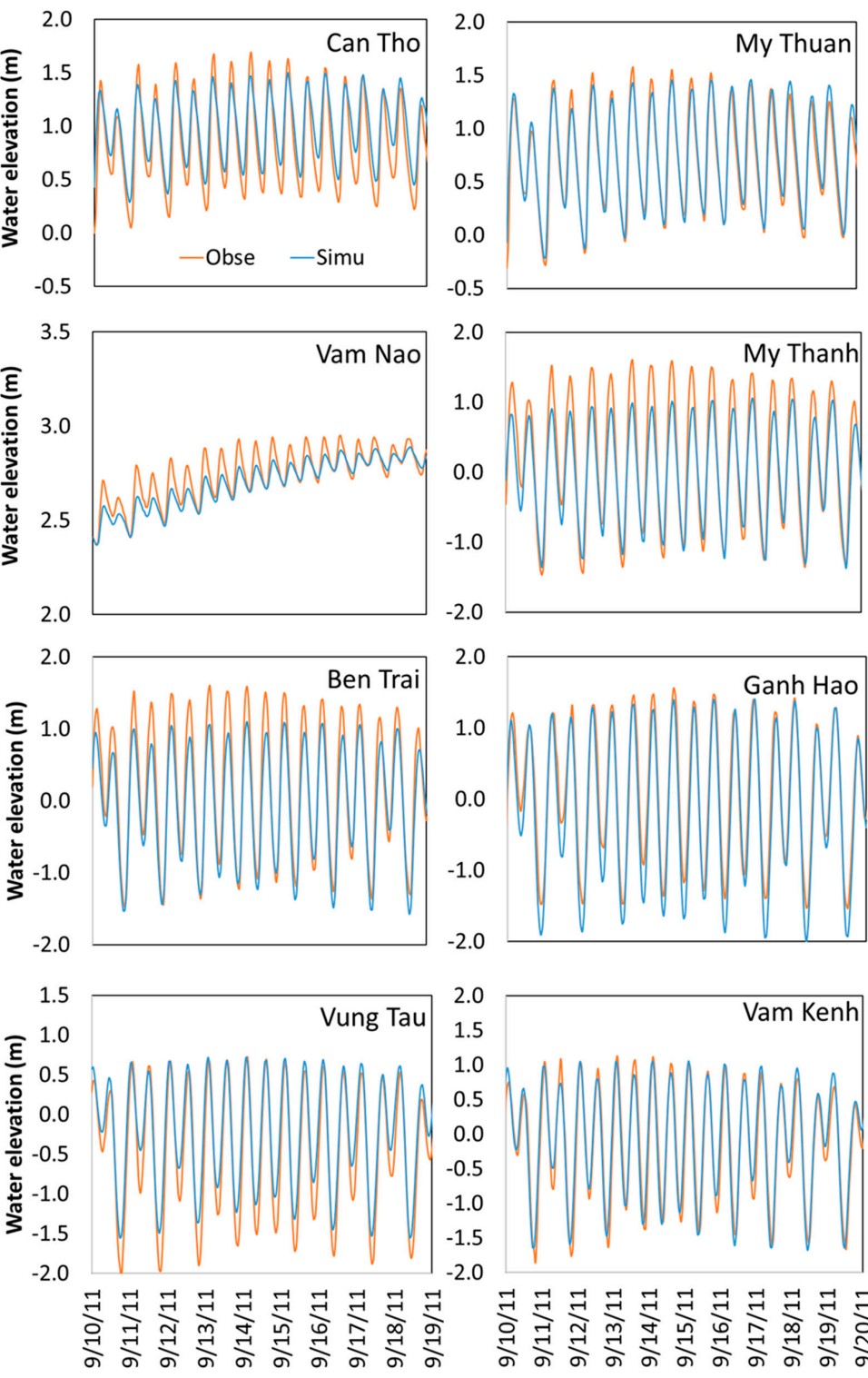

**Figure 6.** Model calibration. The red lines present the measured data, and the blue lines present the simulated one.

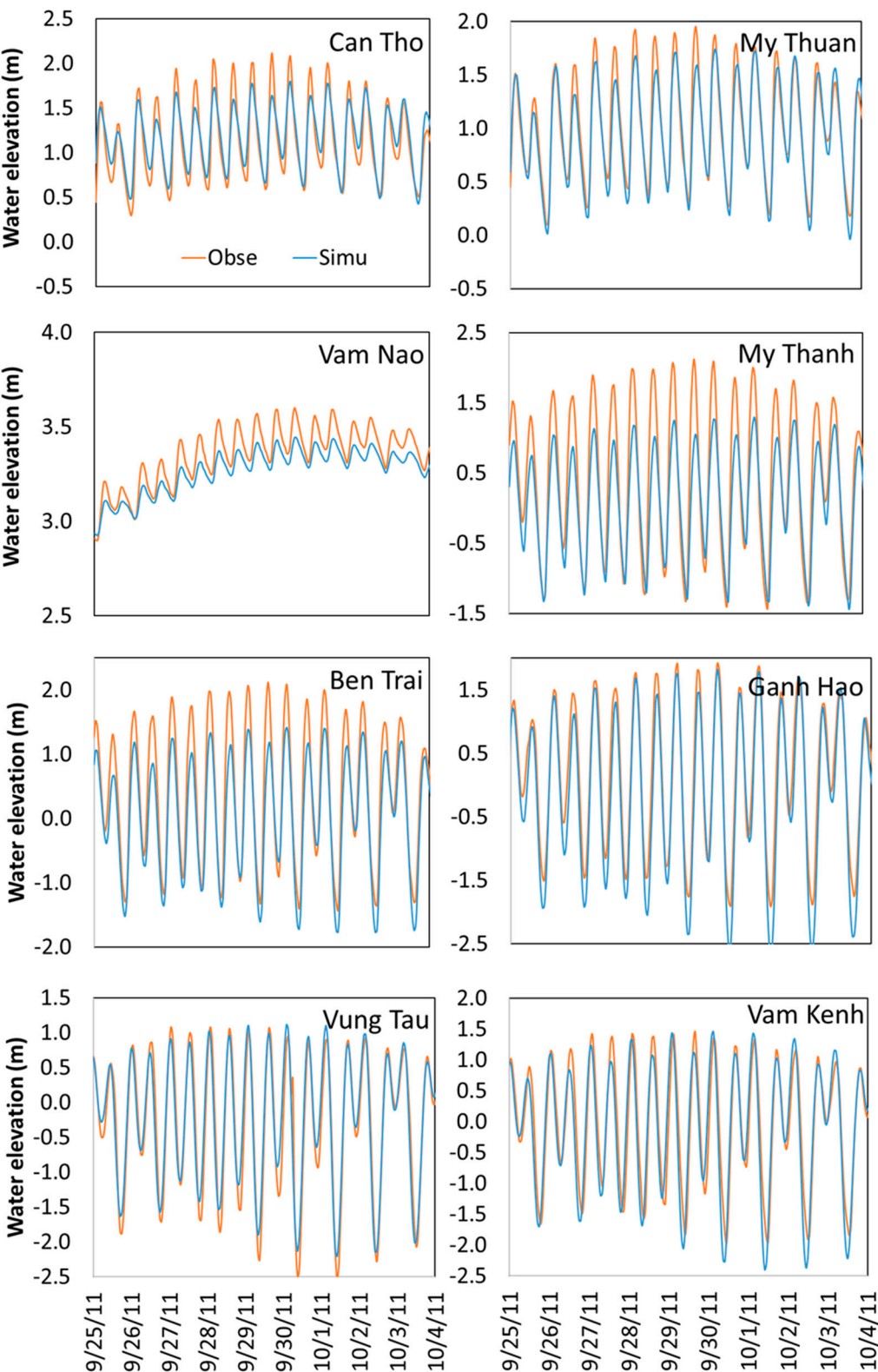

**Figure 7.** Model validation. The red lines present the measured data, and the blue lines present the simulated one.

**Table 4.** Hydrodynamic model validation.

| No. | Station | Calibration 10–19 September 2011 | | | Validation 25 September–4 October 2011 | | |
|---|---|---|---|---|---|---|---|
| | | NSE (%) | MAE (m) | RMSE (m) | NSE (%) | MAE (m) | RMSE (m) |
| 1 | Can Tho | 74.69 | 0.178 | 0.203 | 87.48 | 0.136 | 0.163 |
| 2 | My Thuan | 82.95 | 0.830 | 0.273 | 92.52 | 1.116 | 0.303 |
| 3 | Vam Nao | 74.04 | 0.004 | 0.131 | 60.29 | 0.008 | 0.143 |
| 4 | My Thanh | 87.14 | 0.263 | 0.316 | 76.99 | 0.391 | 0.487 |
| 5 | Ben Trai | 79.98 | 0.249 | 0.292 | 82.16 | 0.360 | 0.429 |
| 6 | Ganh Hao | 90.83 | 1.472 | 0.272 | 88.49 | 1.726 | 0.366 |
| 7 | Vung Tau | 87.26 | 0.225 | 0.275 | 93.04 | 0.192 | 0.249 |
| 8 | Vam Kenh | 92.97 | 0.175 | 0.211 | 84.96 | 0.292 | 0.365 |

The NSE is defined as follows:

$$NSE = 1 - \frac{\sum_{i=1}^{n}\left(H_{obs,i} - H_{simu,i}\right)^2}{\sum_{i=1}^{n}\left(H_{obs,i} - \overline{H_{obs}}\right)^2} \tag{5}$$

where $\overline{H_{obs}}$ is the mean value of observed water depths, $H_{obs,i}$ is the observed water depth at time $t = i\,\Delta t$ and $H_{simu,i}$ is the numerically simulated water depth at time $t = i\,\Delta t$, with $n$ being the total number of time steps.

The MAE and RMSE criterions are applied to measure the absolute differences between simulations and observations.

$$MAE = \frac{\sum_{i=1}^{n}\left|H_{obs,i} - H_{simu,i}\right|}{n} \tag{6}$$

$$MSE = \sqrt{\frac{\sum_{i=1}^{n}\left(H_{obs,i} - H_{simu,i}\right)^2}{n}} \tag{7}$$

The statistical matches are reported in Table 4, with the parameter values presented in Table 3. The numerical simulations show good agreement with the observed data in almost all stations, except the Vam Nao station, where the Tien and Hau Rivers join, with very specific impacts on the local bathymetry due to sand mining [14] and flow redistribution between the Tien and Hau Rivers [61].

The calibration and validation results show some discrepancies, in particular, some noticeable underestimation of water elevation in the stations of Vam Nao, My Thanh and Ben Trai stations. As presented above, the VMD is a wide and complex domain, i.e., a low-lying area where small topography effects interact with meteo-hydrological regimes and tidal forcings. Moreover, the VMD is clearly modified by ongoing anthropogenic stresses and climate changes. For these reasons, it is virtually impossible to capture all the modifications. An exact simulation of the current status of the VMD is not the real objective, but the model is designed to help policymakers and stakeholders in their understanding of the hydrodynamics of the VMD. The model has tried to integrate main drivers and utilize robust input data sources. The most underestimated simulation is found in Vam Nao, where the morphological conditions were significantly altered and severe incision and erosion occurred as reported by [46,62]. Excessive sand mining [4,62], land-use changes [63] and over-exploitation of groundwater [22] activities induced riverbed lowering and instability in the LMB, especially in the VMD [64]. All these supplemental causes contribute to the underestimation of water elevations and model uncertainties.

## 5. Results

Flooding in the VMD usually occurs in two regimes: (1) upstream floods, driven by the hydrological forcing inland, and (2) tidal-induced floods, triggered by the tidal flows from the East and West Seas [7]. The period 25–30 October 2011 is selected to simulate the extreme flood dynamics in the VMD because the maximum water level in 2011 was higher than that of the historic flood event in 2000 [6,64]. For this event, a late flood peak (in October 2011) coincided with a spring tide period [19], which caused the highest water level observed so far in the VMD.

### 5.1. Inundation Level and Area

Figure 8 shows that the water level in the upper VMD is always higher than in the downstream parts, especially in the PoR and LXQ, which are considered to be two flood-prone areas of the VMD. The water level in the PoR is higher than 1.8 m, while the water level in the LXQ is higher than 1.0 m, which redistributes the flow in the VMD by discharging floodwater into the Gulf of Thailand to protect local residents and crops. Thanks to the multi-channel network, inundation areas between Tien and Hau Rivers with water levels higher than 1.0 m are limited to the upstream of Can Tho and My Thuan. This confirms that the multi-channel network does not influence both branches through the hydraulic links in the channel system.

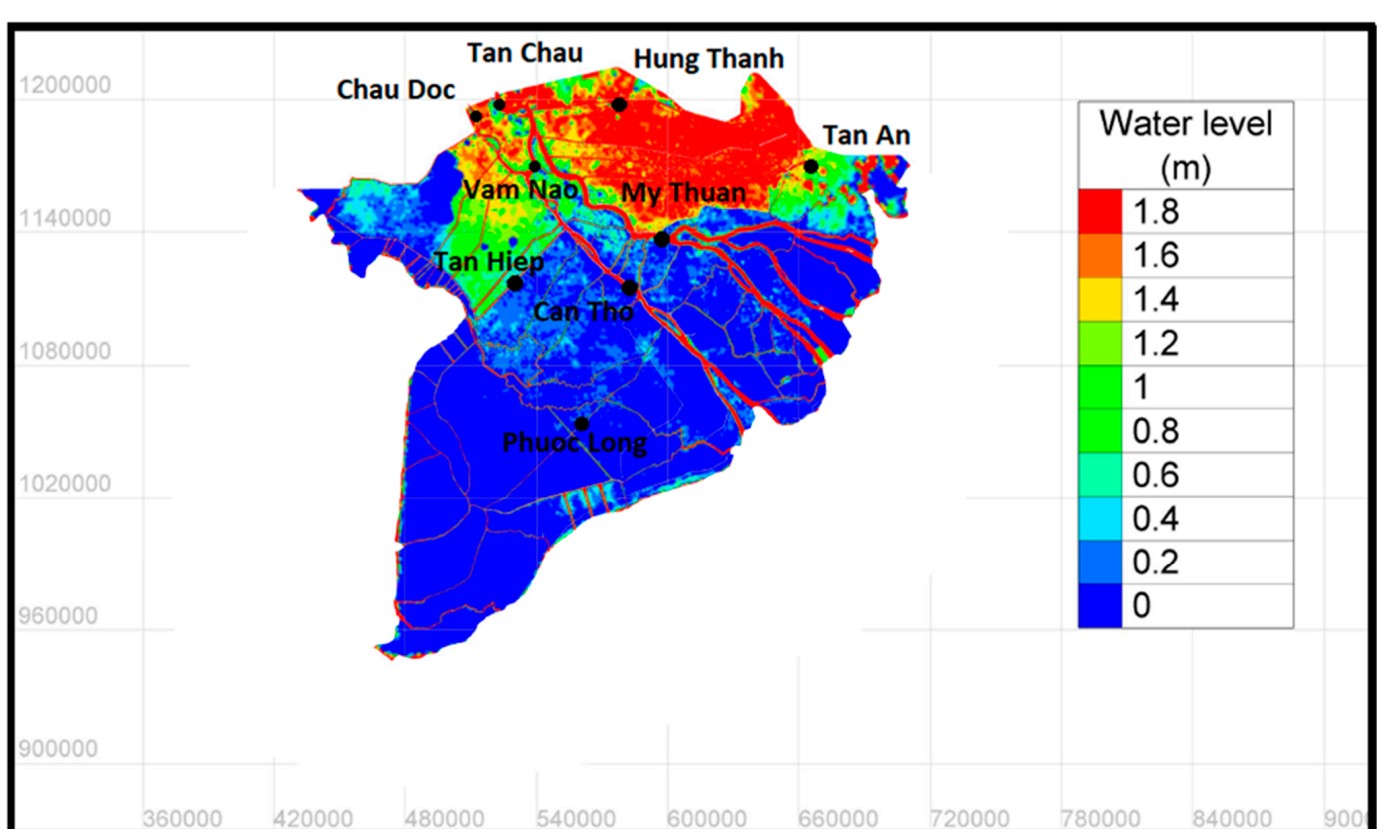

**Figure 8.** Inundation area and water level in the VMD in the Scenario S1 on 30 October 2011.

In this paper, inundation areas are defined as having a water level of 0.5 m or more inland. This is considered to be a threshold value beyond which there can be high property damages and no cultivation [7,37]. The total inundation area in Scenario S1 is 11.710 km$^2$, accounting for 29% of the whole VMD area, which agrees with the studies by [16,42].

### 5.2. Flow Velocity

To assess the effects of the multi-channel network on the flow dynamics of the VMD, the hourly flow velocity is extracted at the stations in two main rivers and their channel systems (locations of the stations are shown in Figure 8).

The flow velocity at the upstream station of the Hau River (Chau Doc) is rather constant with a cross-section average value of 0.75 m/s (Figure 9a). At the Vam Nao confluence, the flow velocity increases to 0.82 m/s because of the flow balance mechanism between the Tien River and Hau River [14]. In Can Tho, under the influence of the tidal regimes, the flow velocity oscillates between 0.46 and 1.2 m/s, with a mean value of 0.8 m/s. Two time series extracted from the channels of the Hau River at the Tan Hiep station (33 km from the main river) and the Phuoc Long station (88 km from the main river) show mean values of 0.5 m/s and 0.3 m/s, respectively (Figure 9a).

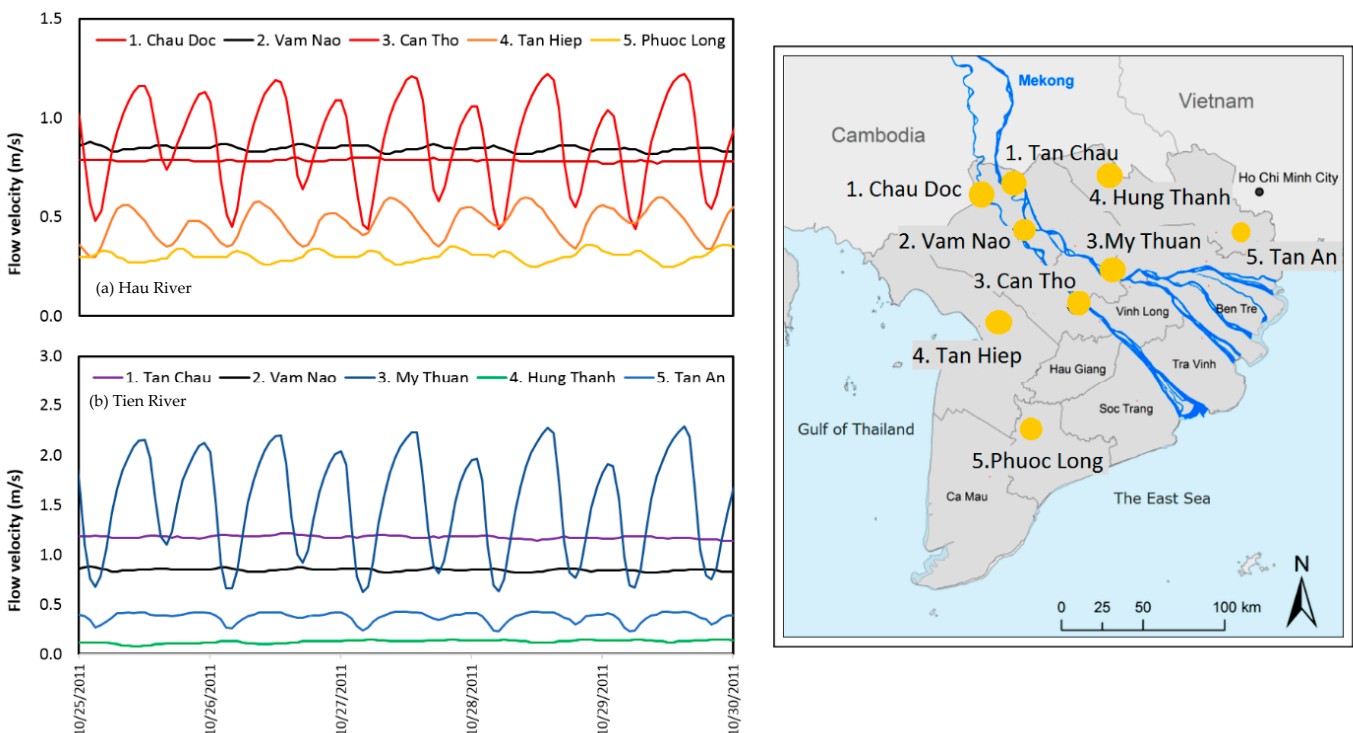

**Figure 9.** Flow velocity in Hau River (**a**) and Tien River (**b**) from the upstream to downstream and their locations in the VMD.

Figure 9b presents the flow velocity in the upstream station of the Tien River (Tan Chau), with a relatively constant value of 1.16 m/s. The fluctuation of the flow velocity in My Thuan is higher than in Can Tho, ranging from 0.66 to 2.24 m/s, with a mean value of 1.45 m/s. Because the influence of the semidiurnal tidal amplitude from the East Sea on the Tien River is stronger than the diurnal tide from the Gulf of Thailand [28,34], velocities in the Hung Thanh station (35 km from the main river) and the Tan An station (25 km from the main river) exhibit mean values of 0.1 m/s and 0.3 m/s, respectively.

Generally, the flow velocity in the channel system decreases with the distance from the main rivers. In the upper part of the VMD (Chau Doc, Tan Chau and Vam Nao), it is dominantly affected by the river flow, whereas the middle part (Can Tho and My Thuan) is under the effects of tidal propagation.

## 6. Discussion

### 6.1. Impacts of the Multi-Channel Network

Scenario S2 (without channels, see Figure 5) was run and compared with results from Scenario S1 (rivers and multi-channel networks, see Figure 4) to assess the impacts of the current multi-channel networks on the flow dynamics in the VMD.

#### 6.1.1. Inundation Level and Area

In Scenario S2, the average inundation level in the PoR and LXQ is approximately 1.0 m and reaches up to 2.0 m in some small areas (see Figure 10), whereas the results from Scenario S1 (the full network scenario) show the maximum water level of the PoR can reach 4.0 m. The total inundation area with water levels of >0.5 m in Scenario 2 (without channels) reaches 25.918 km², accounting for 65% of the VMD area in comparison with 29% for Scenario S1 (Figure 10). The lower water levels and larger inundation areas from Scenario S2 indicate that the channel system is a substantial driver of the hydraulic scheme in the VMD. It works efficiently to drain flood waters from the LXQ and the PoR to the Gulf of Thailand and to the Vam Co River, before discharging to the East Sea, respectively [62].

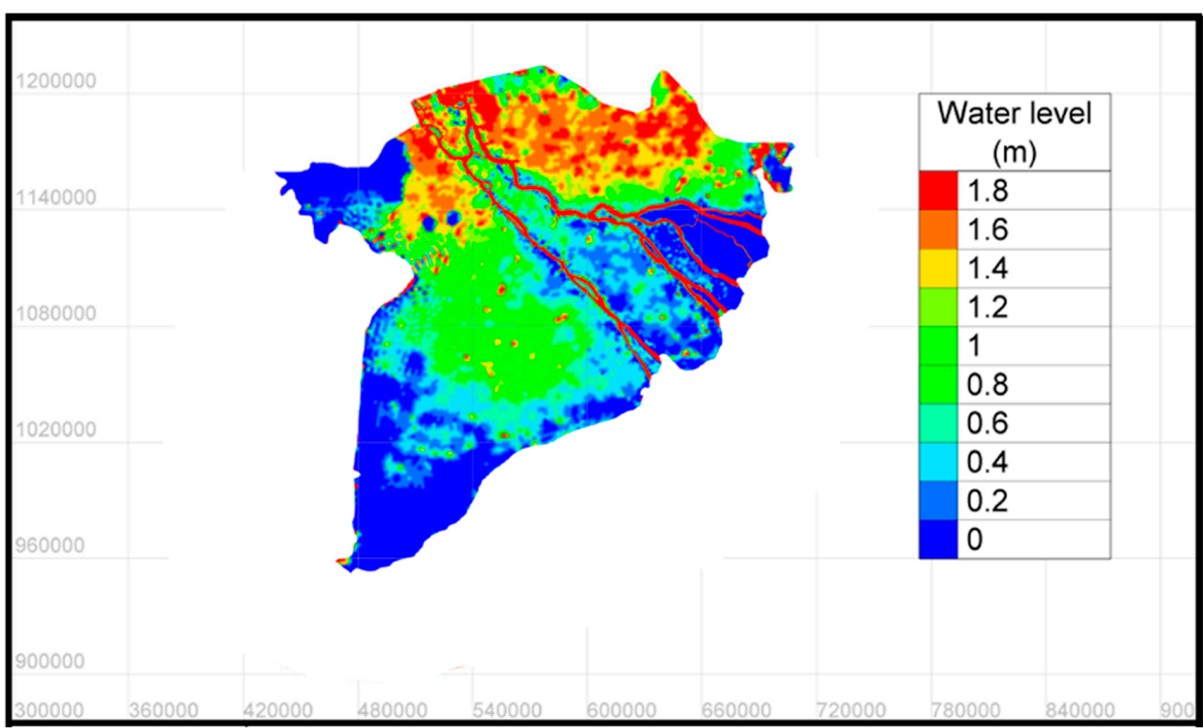

**Figure 10.** Inundation area and water level in the VMD in Scenario S2 (without channels).

#### 6.1.2. Water Elevation and Discharge at Can Tho and My Thuan Stations

Two main stations (Can Tho and My Thuan) were selected to reevaluate the impacts of the complex channel system on the hydrodynamics in the VMD. These stations correspond to two important cities in this region (i.e., Can Tho and My Thuan), with a high-density population, intensive agricultural production and key economic hotspots.

Water elevation increased from 0.11 to 0.43 m in Can Tho and from 0.18 to 0.45 m in My Thuan in Scenario S1 and Scenario S2, respectively. Considering Scenario S2 (without channels), the discharge at Can Tho station increased slightly (about 2000 m³/s); but discharge at My Thuan station increased noticeably by about 5400 m³/s and reached the maximum value of approximately 24,000 m³/s.

At these stations, the tidal signal is clearly visible even during the highest flood event [11,32]. However, the multi-channel systems have reduced the tidal effects and variation of floodwater elevation in both rivers (see Figure 11, Scenario S1, blue curves).

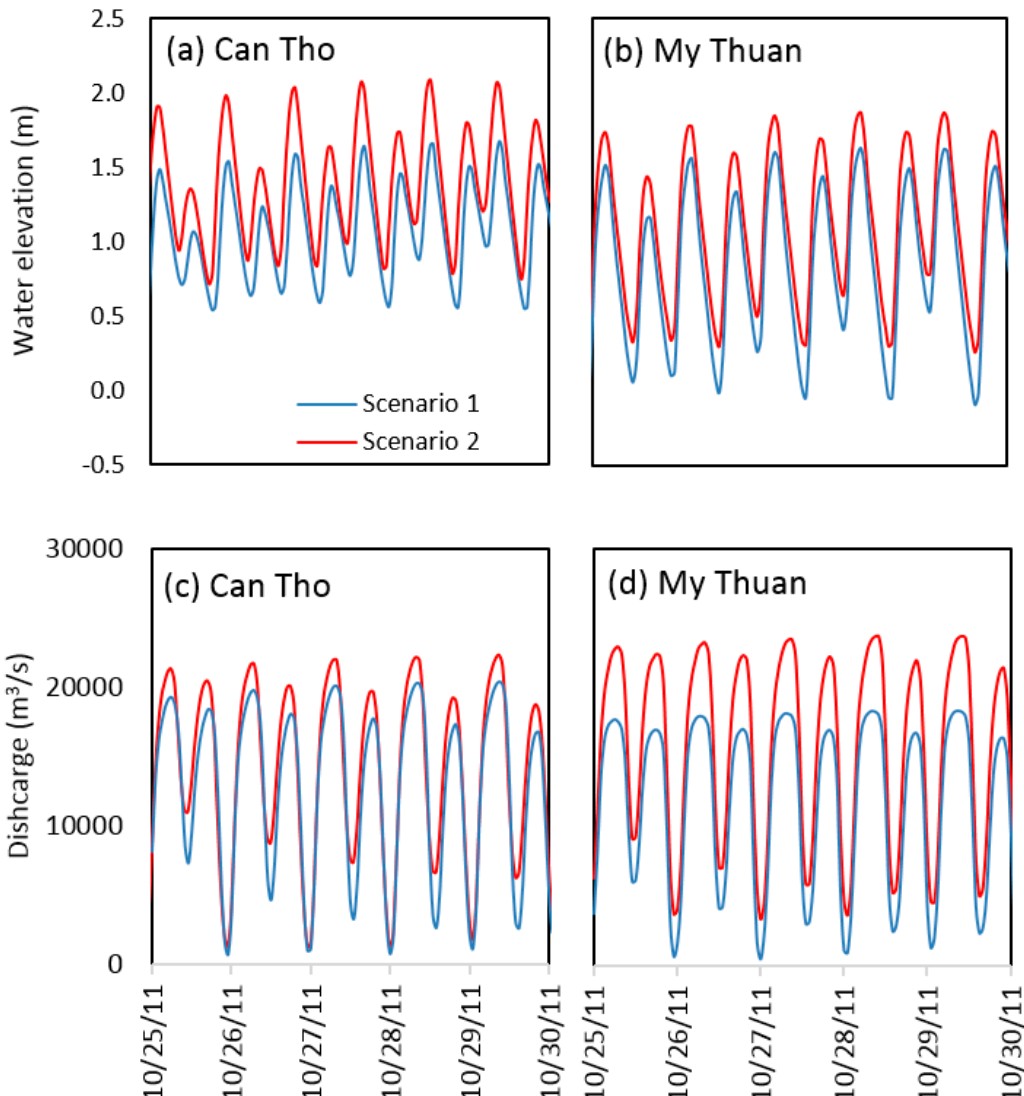

**Figure 11.** Water elevation and discharge at Can Tho (**a,c**) and My Thuan (**b,d**) for two scenarios.

The historical flood year 2011 caused a large inundation area in the VMD (see Figure 10 and [38]). The maximum simulated floodwater elevation at the main stations in both scenarios was compared with the Vietnam flooding alarm elevation No. 1 (Alarm No. 1) to define the flooded areas and assess the impacts of the channel network on flood protection.

Water elevations at almost all stations in the upper VMD were higher than Alarm No. 1 in both scenarios because the upstream flood had a dominant influence on the flow dynamics, especially the PoR and the LXQ. These regions store water in the wet season and supply water in the dry season for the whole VMD. The water elevation in the coastal area was generally lower than Alarm No. 1 (Phung Hiep, Tra Vinh, My Tho and Dai Ngai), except for Vam Kenh and Hoa Binh.

Water elevations in Scenario S1 are lower than those in Scenario S2 because the channel network works as a water conveyor from the VMD to the seas and mitigates flooding. Thus, the water levels in Can Tho and My Thuan in Scenario S1 are approximately at the Alarm No. 1 level. The intercomparison of scenarios, which are synthesized in Figure 12, underlines the effect of the multi-channel network on flood reduction, especially in the coastal areas and key cities of the VMD.

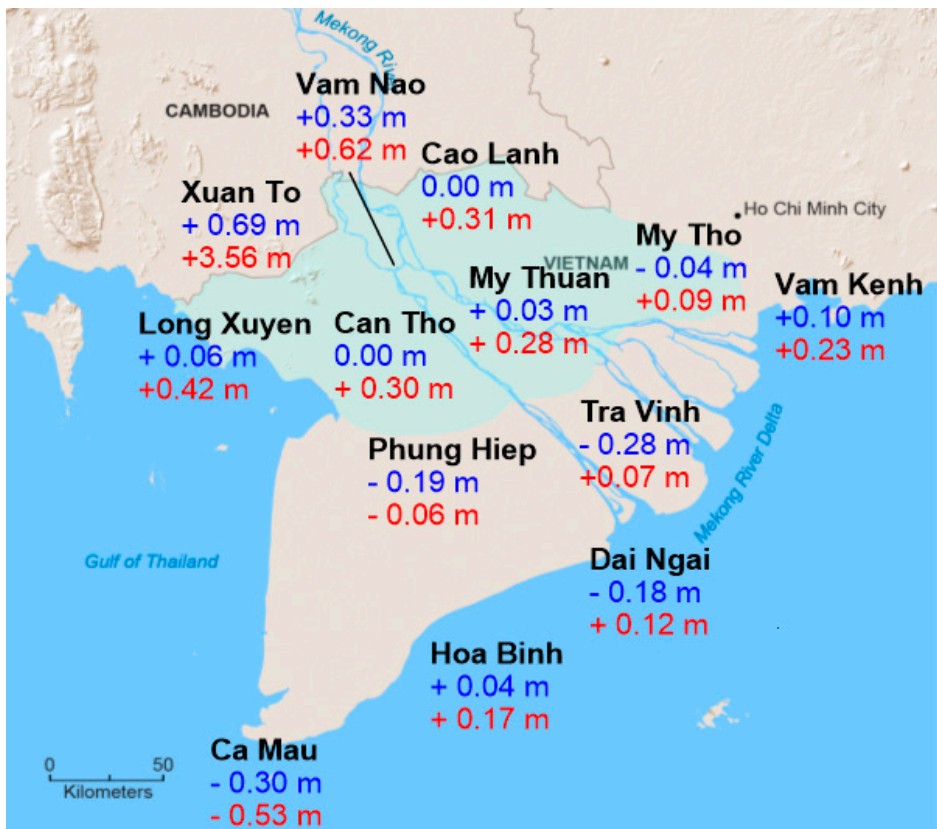

**Figure 12.** The highest water elevations at main stations in comparison with the Vietnam alarm elevation No. 1. Blue texts show the results from Scenario S1 (with channels); red texts present the results from Scenario S2 (without channels). Positive values mean the water elevation at the station higher than the Vietnam alarm elevation No. 1; negative values mean the water elevation lower than the Vietnam alarm elevation No. 1. The light blue area delimits the inundation area common to both scenarios.

### 6.2. Impacts of Sea–Level Rise

Four additional scenarios (S3, S4, S5, S6), as introduced in Table 2, were simulated to evaluate the impacts of RSLR on the flow patterns of the VMD, with visions to 2050 (+30 cm) and 2100 (+100 cm).

#### 6.2.1. Water Elevations and Discharges

RSLR will likely pose additional pressures on the development of this region [5,47]. It may propagate flooding and cause higher water levels and discharge in the hotspot cities of the VMD, namely, Can Tho and My Thuan. The results of these scenarios are provided in Figures 13 and 14 and Table 5. With the multi-channel network, water levels in Can Tho increase by 0.28 m and 0.55 m for the projected RSLR of 30 cm (in 2050) and 100 cm (in 2100), respectively. Water levels in My Thuan increase by 0.29 m and 0.55 m for the same conditions. The discharge also increases to 6004 $m^3$/s and 7479 $m^3$/s in Can Tho and to 6844 $m^3$/s and 9276 $m^3$/s in My Thuan, respectively, because water is added to the main rivers from the floodplain and RSLR pushes ocean tides upstream. Without the multi-channel network (S4 and S6), the water level in Can Tho increases by 0.24 m and 0.47 m, respectively, while the increases in My Thuan are 0.30 m and 0.63 m, respectively. The discharges increase to 4530 $m^3$/s and 6066 $m^3$/s in Can Tho and to 2985 $m^3$/s and 5418 $m^3$/s in My Thuan, respectively.

The increase of water levels and discharges without the multi-channel network (Scenarios S2, S4 and S6) are greater than those with the multi-channel network (Scenarios S1, S3 and S5) because the tidal regimes directly influence the flow regimes of the Hau River

and the Tien River (see Figures 13 and 14). The tidal amplitudes of Can Tho and My Thuan reach the maximum values of 2.0 m and 2.1 m, which is higher than the values of 1.0 m and 1.7 m in the scenarios with the multi-channel network. The multi-channel network buffers hydrodynamics in the floodplain and may lessen the floodings and associated damages.

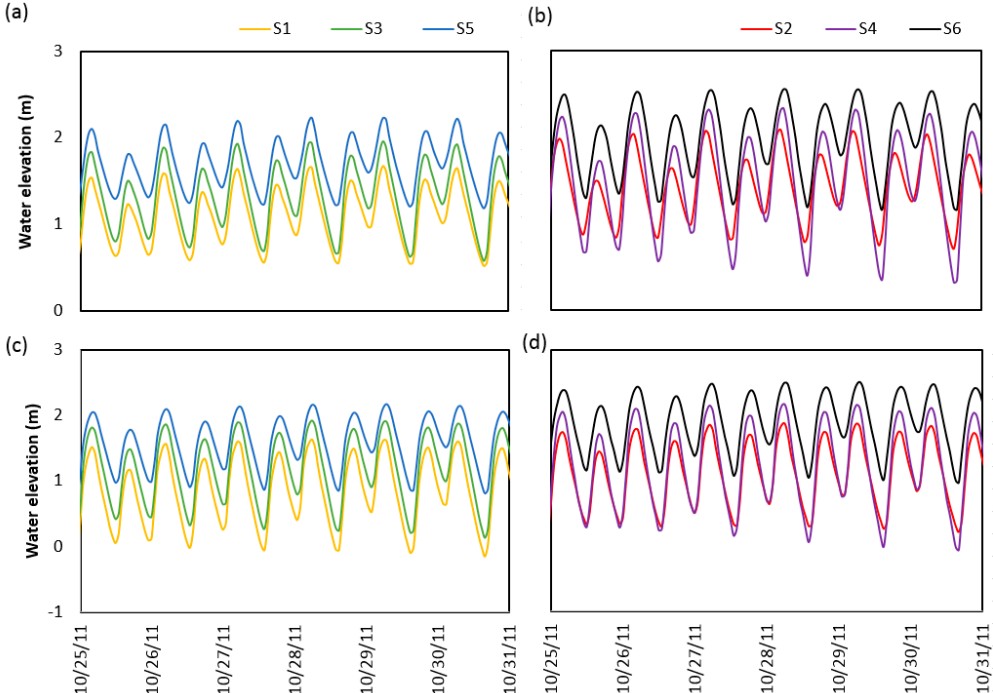

**Figure 13.** Water elevation in Can Tho (**a,b**) and My Thuan (**c,d**) stations corresponding with Scenarios S1, S3 and S5 and Scenarios S2, S4 and S6.

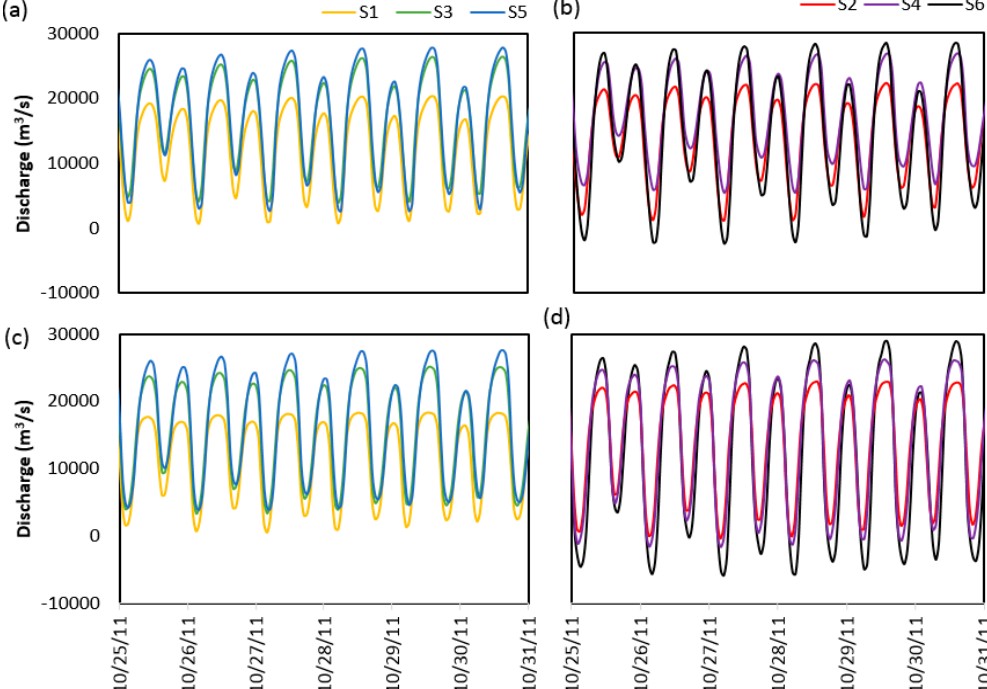

**Figure 14.** Discharge in Can Tho (**a,b**) and My Thuan (**c,d**) stations corresponding with Scenarios S1, S3 and S5 and Scenarios S2, S4 and S6.

**Table 5.** Summary of results from the 6 scenarios considered in this study.

| Parameter | S1 | S2 | S3 | S4 | S5 | S6 |
|---|---|---|---|---|---|---|
| Max. water level at Can Tho (m) | 1.68 | 2.09 | 1.96 | 2.33 | 2.23 | 2.56 |
| Max. water level at My Thuan (m) | 1.62 | 1.87 | 1.91 | 2.17 | 2.17 | 2.50 |
| Max. discharge at Can Tho (m$^3$/s) | 20,396 | 22,358 | 26,400 | 26,888 | 27,875 | 28,424 |
| Max. discharge at My Thuan (m$^3$/s) | 18,348 | 23,765 | 25,192 | 26,750 | 27,624 | 29,183 |
| Inundation area (km$^2$) | 11,710 | 25,918 | 12,750 | 27,617 | 20,385 | 34,199 |
| Percentage in the VMD | 29% | 65% | 32% | 69% | 51% | 85% |

### 6.2.2. Inundation Area

Our modelling results suggest that hydrological regimes of the coastal and estuarine parts of the VMD would be much more heavily affected by the RSLR than in the upper part (see Figure 15). The largest inundation area in the VMD, calculated in Scenario S1, is 11,710 km$^2$, corresponding to 29% of the total VMD area. This area increases to 12,750 km$^2$ in 2050 (Scenario S3, RSLR + 30 cm) and 20,385 km$^2$ in 2100 (Scenario S5, RSLR + 100 cm) towards the coastal part of the VMD [37,47], corresponding to 32% and 51% of the total area, respectively. However, without the effects of the multi-channel network, the maximum inundation area by floodwater would have risen up to 27,617 km$^2$ (Scenario S4, RSLR + 30 cm) and 34,199 km$^2$ (Scenario S6, RSLR + 100 cm), accounting for 69% and 85% of the total area, respectively. The study of [22] also confirms this threat to the VMD at the end of this century, with higher risks of flooding than previously assessed.

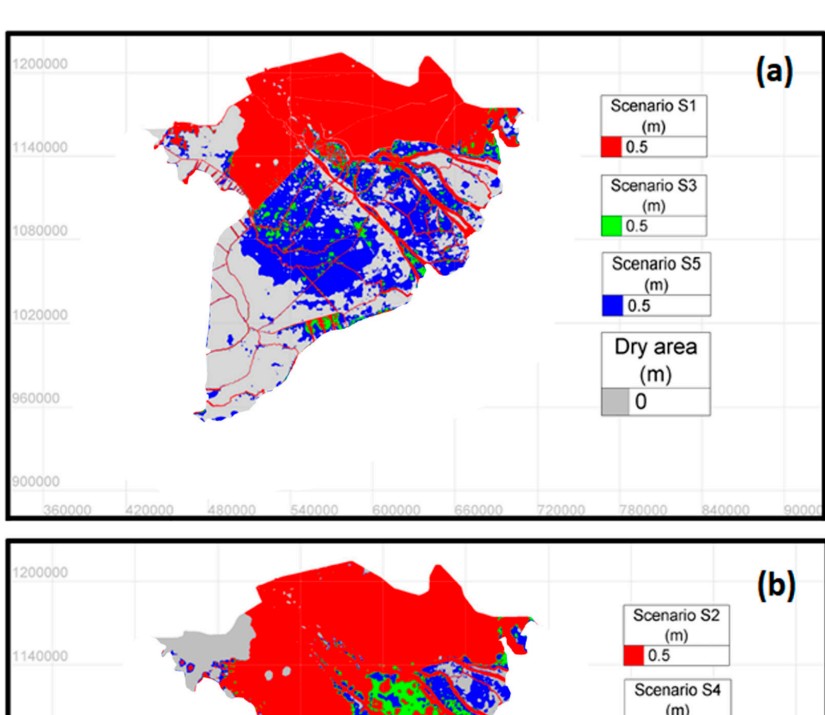

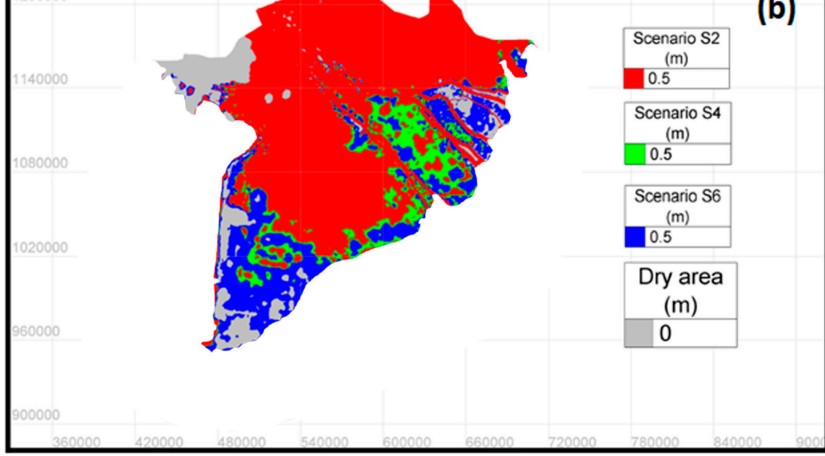

**Figure 15.** Inundation areas with water level > 0.5 m in different scenarios. (**a**) VMD with the multi-channel network; (**b**) VMD without the channel network.

## 7. Conclusions

This study confirms the capacity of a full 2D model to simulate flood dynamics in a large-scale domain under many drivers such as man-made structures and natural conditions (flood events, tidal motions and wind stress). Our study provides information regarding the vulnerability of the VMD to flooding nowadays and in the future. This spatial prediction may be useful for the local residents and stakeholders to adopt proper prevention measures.

The results also shed light on the link between hydrodynamic alterations and human interventions in the VMD. The inundation area in the VMD in 2011 was calculated to be approximately 12,000 km$^2$, with water levels ranging from 0.5 to 4.0 m. The historical construction of the channel network reduced flooding areas, helping local communities to lessen flood-driven damage and increased agricultural production areas. Without the multi-channel network, the inundation area would have been larger, extending to approximately 26,000 km$^2$ and causing serious damages and human suffering.

RSLR will also likely bring additional pressures on the water-related safety and sustainability of this region, as pointed out by many authors [18,23,37,65]. The cumulative impact of subsidence and sea-level rise will increase risks of inundation both in space and frequency. The maximum inundation area could reach about 35,000 km$^2$, accounting for 85% of the whole VMD area in 2100. Thus, large scale measures to mitigate or counterbalance the effects of RSLR and other drivers such as land subsidence and storm surges are now essential to maintain agricultural cultivation and minimize human suffering in the VMD.

In the light of the results of this numerical study, we can assume that a sustainable development of the multi-channel network will bring an optimal regulation of water cycles and sediment transport patterns in the VMD. It could contribute to improve agricultural production and livelihoods in the region, even in the extreme conditions of climate change and sea-level rise. For future studies, the sediment distribution pattern in the rain season and the flow distributions and saltwater intrusion in the dry season should be addressed to anticipate this additional threat.

**Author Contributions:** Conceptualization, data curation, formal analysis, modelling, writing, H.-A.L.; data collection, resources, modelling setup, T.N.; methodology, visualization, review, editing, N.G.; funding acquisition, project administration, review, S.S.-F.; review and supervision, E.D. All authors have read and agreed to the published version of the manuscript.

**Funding:** This research received no external funding.

**Acknowledgments:** Eric Deleersnijder and Sandra Soares-Frazão are honorary research associates with the Belgian Fund for Scientific Research (F.R.S.-FNRS).

**Conflicts of Interest:** The authors declare no conflict of interest. The funders had no role in the design of the study; in the collection, analyses or interpretation of data; in the writing of the manuscript or in the decision to publish the results.

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
