# Peer review of "The Multi-Channel System of the Vietnamese Mekong Delta: Impacts on the Flow Dynamics under Relative Sea-Level Rise Scenarios"

_water, doi:10.3390/w15203597_

Round 1

Reviewer 1 Report

General comments:

1. Please add to the introduction in more detail about the previously articles which conducted to modeling of Sea level in the Mekong Delta, which models, what results was obtained. Add several links to case studies about the extreme floods in the Mekong Delta.

Please add several links to papers about the global sea level increase and increase estimates in your investigating area. 

2. What land and ocean bathymetry was used? What it is quality and spatial resolution of bathymetry and topography ? We know that gebco and etopo has +-10 m in shallow water and it is very bad for the models with 100 m space resolution. Land topography from ASTER and SRTM - very bad when the floods calculatiing. It is a principal question.

3. The storm surges was not included in the model.  What the possible errors you can get from this factor? What may be culative effects? Please add comments. Please add some direct links to estimations of amplitude of strom surges in your region from literature.

4. fig 7 - please explain the underestimating in more part of the model results.

Minor commetns:

1. Fig 1 - please show the map with geografical coorditnates and show ocean and seas.

2. Please add to introduction  othe cases of using TELEMAC model in different regions. Add the model quality estimation results.

3. Fig 3 - is it not dephs > 7 m here? please check color bar and ocean depth  isolines.

4. "Hourly wind data at 10 m from NCEP NOAA  (https://www.ncep.noaa.gov/) are used in the simulation" - which one of wind  analysis/reanalysis you use? which spatial resolution? Please add this information.

5. In which software was developed unstructured computation mesh? Please add this information.

6. Which water leve sensors used in field measurements? What is it accuracy and time resolution? Please add this information.

Author Response

Dear Reviewer 1,
Thank you for giving us the opportunity to submit a revised draft of our manuscript titled The multi-channel system of the Vietnamese Mekong Delta: impacts on the flow dynamics under relative sea level rise scenarios to MDPI Water. We appreciate the time and effort that you and  Reviewer 2 have dedicated to providing your valuable feedback on my manuscript. We are grateful to the reviewers for your insightful comments on our paper. We have been able to incorporate changes to reflect most of the suggestions provided by the reviewers. We have highlighted the changes within the manuscript.

Kindly find our point-by-point responses to your comments and concerns.

Best regards,

Hoang Anh Le

On behalf of co-authors

Reviewer 2 Report

I sincerely appreciate the dedication and effort invested in preparing the manuscript. The research presented in this work utilizes 2D TELEMAC model to address a highly significant global concern pertaining to hydrodynamic and sediment transport in large river deltas in the world under climate change. I have several comments and suggestions to improve the clarity and overall quality of the manuscript:

1- Introduction: this section can be improved by stressing the novelty of the present research by showing what is the main difference between the simulation in this study as compared to other studies in the Vietnamese Mekong Delta (VMD). In my point of view, there have been several studies on simulating the hydrodynamic conditions of river network in the VMD using 2D numerical models.

- - The headings of Section 3 should be corrected as “Model set-up”

3- The equations should be the same font as the text in the MS. I recommend the authors to use an Equation editor tool to prepare equations.

4- Table 1: The river network and cross-section data are updated to 2010. Is this data suitable for the simulation?

5- In the Introduction section, the authors stressed that the new point of the present study is using 2D model TELEMAC to simulate “overland flow and water exchange between irrigation compartments” and this idea was then depicted in Figure 2 of the MS. However, from Lines 195 to 212, only the two main rivers and the eight estuaries are modeled, and the tertiary and quaternary channels are neglected. Is this affected the purposed of simulating the real conditions of the VMD as stated in Scenario 1?

Author Response

Dear Reviewer 2,
Thank you for giving us the opportunity to submit a revised draft of our manuscript titled The multi-channel system of the Vietnamese Mekong Delta: impacts on the flow dynamics under relative sea level rise scenarios to MDPI Water. We appreciate the time and effort that you and  Reviewer 1 have dedicated to providing your valuable feedback on my manuscript. We are grateful to the reviewers for your insightful comments on our paper. We have been able to incorporate changes to reflect most of the suggestions provided by the reviewers. We have highlighted the changes within the manuscript.

Kindly find our point-by-point responses to your comments and concerns.

Best regards,

Hoang Anh Le

On behalf of co-authors

Round 2

Reviewer 1 Report

Thank you for your work, but several your answers didn't satisfy me.

1 Comment 2: What land and ocean bathymetry was used? What it is quality and spatial resolution of bathymetry and topography ? We know that gebco and etopo has +-10 m in shallow water and it is very bad for the models with 100 m space resolution. Land topography from ASTER and SRTM - very bad when the floods calculatiing. It is a principal question.

Response 2: The domain and bathymetry of the Vietnam Mekong Delta, which was developed from several sources (HoChiMinh city University of Technology, Vietnam, the Southern Institute for Water Resources Research, and Vietnam MRC). These data were derived from the 100 m x 100 m grid resolution Digital Elevation Map (DEM), with reference to the Hondau datum (the Vietnamese official benchmark system, identical to mean sea level).

In your answer I didn't see the quality/accuracy and spatial resolution of used bathymetry and topography. What was original sources for DEM? If it has a bad quality - your results will be incorrect. And you  need to improove your model implementation.

It is a principal question to reject or accept this paper!!!

2 Comment 10: Which water leve sensors used in field measurements? What is it accuracy and time resolution? Please add this information.

Response 10: Thank you for pointing this out. We did not use the field measurement data for this study.

line 305-306 "The water levels were evaluated at eight measurement stations (Figure 1a, Figure 6, 305 Fig 7 and Table 4), which are key stations with the highest data accuracy..." Observation presented on figure 6

I must to repeat the question: What is it accuracy and time resolution? Please add this information to Section 4.

Author Response

Dear Reviewer 1,
Thank you for giving us the opportunity to resubmit our revised manuscript titled The multi-channel system of the Vietnamese Mekong Delta: impacts on the flow dynamics under relative sea level rise scenarios to MDPI Water. We appreciate the time and effort that you have dedicated to providing your valuable feedback on my manuscript. We have clarified your comments in our point-by-point responses.

Kindly find the attachment.

Best regards,

Hoang Anh Le

On behalf of co-authors

Round 3

Reviewer 1 Report

1. Please add shortly information about  land and ocean bathymetry (i see it in your ansver, but didn't see in manuscript), it quality (I didn't see it in the manuscript!!) and spatial resolution 

2."Data used for the model calibration and validation is extracted from several official sources"

- Please extract form this "several official sources" information abut te sensors accuracy and time resolution of measurements. it is very important. Please add this information to the text.

Author Response

(The authors gave the same response as above.)
